# Effects of Human Disturbance on Riparian Wetland Landscape Pattern in a Coastal Region

Shiguang Shen [1] , Jie Pu [1], Cong Xu [1], Yuhua Wang [2], Wan Luo [3] and Bo Wen [1],*

1   College of Landscape Architecture, Nanjing Forestry University, Nanjing 210037, China
2   College of Horticulture, Nanjing Agricultural University, Nanjing 210095, China
3   Boyuan Planning and Design Group Co., Ltd., Nanjing 210004, China
*   Correspondence: wenbo2019@njfu.edu.cn

**Abstract:** The wetland ecosystem along a river in a coastal region has great significance in purifying water bodies, regulating climate, and providing habitat for animals and plants. Studying the effects of human disturbances on the landscape patterns of wetlands is of great significance to the protection and management of an ecosystem. This study used Guannan County and Guanyun County, two counties in China that are located on both banks of the Xinyi River as the study area. The spatiotemporal characteristics of the landscape pattern evolution of wetlands and their relationship with human interference from 2009 to 2020 were analyzed by the landscape dynamic rate, landscape conversion matrix, landscape indices, human disturbance index, and the quadratic regression equation. The results showed that: (1) Except for the increase in the area of beach and paddy fields, the area of other landscape types decreased; (2) the changes in wetlands were heterogeneous and showed different trends in different regions; (3) the boundary shape's complexity and the landscape pattern's fragmentation showed a decreasing–increasing trend and the connectivity and the diversity of the landscape decreased; and (4) the human disturbance index increased from 2009 to 2014 and then decreased from 2014 to 2020, declining outward from the places where towns and construction land aggregated. Moreover, there was an inverted U-type relationship with the landscape pattern indices. The findings provide direct, specific, and explicit information and theoretical guidance for the protection of wetlands along the river in the coastal region as well as for the restoration of wetland ecosystem function and the improvement of wetland biodiversity in relevant regions.

**Keywords:** wetlands evolution; human effects; driving forces; nonlinear relation; quadratic regression equation





## 1. Introduction

The riparian area along the river in a coastal region is an integral part of the global biochemical cycle as it carries many runoffs, which can transport a large amount of material into the sea. Here, wetlands play a particularly prominent role in purifying the water for maintaining ecosystem health as they can provide various ecosystem services such as improving the water quality, regulating climate, and mitigating storm surges [1–3]. Additionally, wetlands located in the transition zone between the mainland and the ocean can provide natural living environments for animal and plant life, which generally support higher biodiversity [4,5]. Attention should be paid to the wetlands' protection and utilization in the rivers' riparian area in the coastal regions, especially in the rapid development stage.

Due to the natural and artificial factors, the wetlands have changed drastically over the past decades, and in particular, the wetlands located near human gathering areas have experienced intensive destruction [6,7]. The influence of natural factors on the change in wetlands is mainly reflected in climate change and biological invasion [8,9], and the human disturbance factors mainly include industrialization, urbanization, agriculture reclamation, and so on [10]. Rapid industrialization and urbanization, for instance, accelerate

the exploitation of limited resources, causing environmental pollution problems [11–13]. Moreover, agricultural modernization not only increases crop yields and promotes the development of the rural economy, but also brings widespread pollution from the overuse of fertilizers and pesticides in rural areas [14,15]. As a result, many wetlands have been lost and the fragmentation degree of wetlands has increased. Large areas of wetland are disappearing or converting into other land use types, and those remaining areas become vulnerable to anthropogenic impacts [16]. This change brought about the segmenting of original wetland patches and a decrease in landscape diversity, thereby resulting in degeneration in the ecosystem and its functions [17,18]. Hence, it is necessary to place focus on researching the wetlands in the riparian area along the river into the sea so that the importance of wetlands can be emphasized and the ecological function can also be protected and improved.

The spatial and temporal changes of wetlands have received wide attention, and extensive research has been carried out in recent years. Many methods such as the spatial analysis of GIS, landscape conversion matrix, landscape pattern index, and human disturbance index have been used to research the important influence of the changes in wetlands [19–24]. Among them, the human disturbance index is a concept opposite to the natural disturbance index and has been widely used in ecological evaluation research in agriculture, forestry, landscape, cities, and other fields, especially the impact of human disturbance on wetlands. Although the research on the wetland landscape change and the human disturbance index has made some progress, correlation analysis has been conducted and the nonlinear relationships could be observed but ignored [20]. The nonlinear characteristics are significant to the policymaking of wetland protection. The selection of variables and methods of the previous research provided the basis for the study of wetlands in this study [25,26]. The quadratic regression has been used to analyze the nonlinear characteristics in the fields of environmental conservation, energy consumption, and driving forces of land use change [27–30]. This study introduces the quadratic regression equation to quantitatively investigate the impact of human activities on the landscape pattern. At the same time, the study of wetlands in the riparian area along the river in a coastal region is relatively rare, but it is a topic worth studying when the development and protection are in contradiction. It is necessary to launch specific research on the spatiotemporal evolution processes of the wetlands in these areas.

Xinyi River is one of the main runoffs flowing into the sea in Lianyungang City, Northern Jiangsu. It has a dense river network and widespread distribution of natural and artificial wetlands. However, under the influence of urbanization, industrialization, and agricultural modernization in Jiangsu Province, wetlands in this area have undergone significant changes. Understanding the linkages between wetland conversion to human activities is important. As Guannan County and Guanyun County are located on both banks of the Xinyi River, this study took these two counties as the study area. At the same time, the past decade has been an important period for the social and economic development of the two places, and relevant policies have also had positive or negative impacts on the region, so the period from 2009 to 2020 was selected as the research interval. In conclusion, the study of wetland changes and human interference in this region is of great significance for local wetland protection and ecologically sustainable development. The objectives of the present study are (1) to reveal the characteristics of the wetland landscape pattern changes in the riparian area along the river into the sea in the past 10 years; (2) to analyze the spatio-temporal heterogeneity of human disturbance in this region; and (3) to analyze the nonlinear relationship between human activities and wetland landscape pattern. Moreover, this study provides theoretical support for the protection and restoration of wetlands in the riparian area along the river in a coastal region, which is conducive to the ecological restoration and sustainable development of wetlands in relevant areas.

## 2. Materials and Methods

### 2.1. Study Area

The study area (33°59′ to 34°39′ N, 119°03′ to 119°52′ E) is situated in the northeastern area of Jiangsu Province, which is a major economic province and in the eastern coastal center of mainland China (Figure 1) [31]. This area is located at the intersection of the Yangtze River Economic Belt and the coastal economic belt. Lianyungang is one of the first batches of Chinese coastal cities opening to the outside world. The regional continuity and similarity of the influences need to be counted in the study of the impact of human activities on wetlands in the riparian area along the river in a coastal region. Guannan County and Guanyun County are located on the south and north bank of the Xinyi River, an important river entering the sea in Jiangsu Plain, and the social and economic development and land use patterns of the two regions are similar. It is typical to use these two administrative regions to analyze the impact of human activities on wetlands in the riparian area along the river in a coastal region as a whole. The total area of this region is 2556.82 km$^2$, and the terrain is flat with a slight incline from west to east. As cities have pleasant living environments, they are rich in water resources and possess a vast water area. Furthermore, natural and artificial wetlands are widely distributed in the study area, and the area of ecological wetlands exceeds 10% of the land area in this region [32]. In addition, in this study, wetlands mainly include the six categories of ditch, river, pond, beach, paddy field, and brine pan, accounting for more than 50% of the whole area.

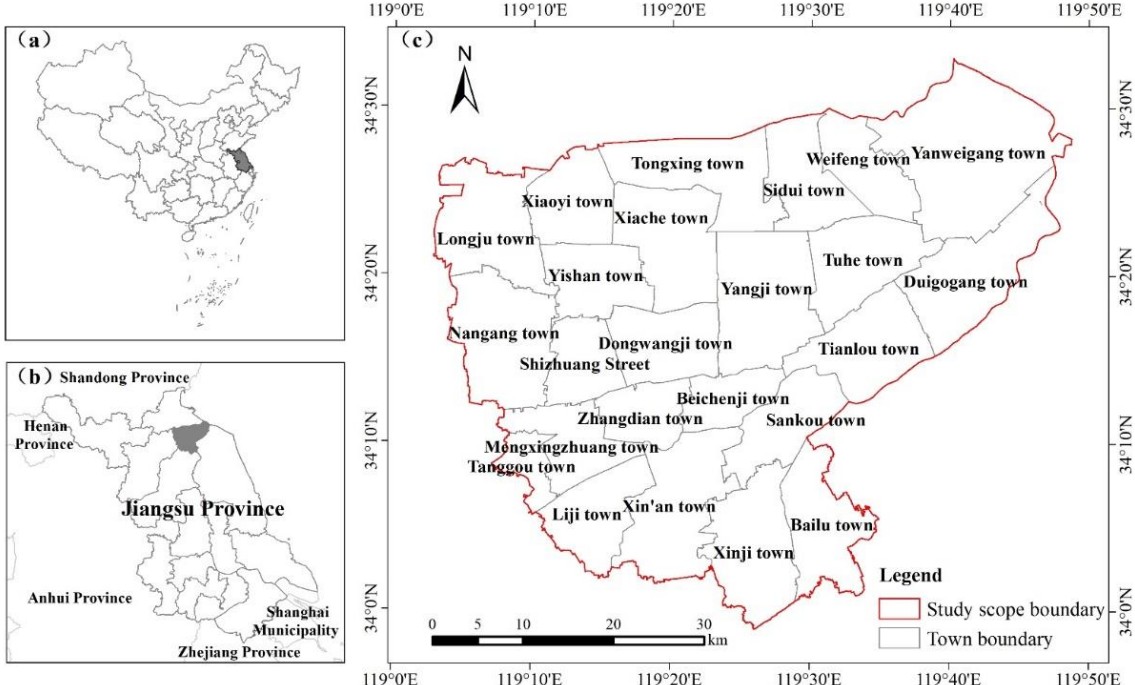

**Figure 1.** Location and extent of the research area. (**a**) Location of Jiangsu Province in China, (**b**) location of the study area in Jiangsu Province, and (**c**) specific location of the study area.

### 2.2. Data Collection

The land use/land cover (LULC) data for 2009, 2014, and 2020 were obtained from Landsat TM images from 2009, and Landsat OLI images from 2014 and 2020 (Table 1). We selected April to July as the study period due to the images of this time having few clouds and the differences in vegetation coverage being small, which improved the classification accuracy. Before LULC data extraction, the remote sensing image preprocessing including geometric, topographic, and radiometric corrections was performed using ENVI 5.3 software [33]. According to the Chinese National standard Current land use classification (GB/T 21010–2017) and the previous studies [6,11,13], 14 types of LULC types were clas-

sified including ditch, river, pond, beach, paddy field, dryland, other arable land, brine pan, construction land, garden, grassland, other unused lands, traffic land, and woodland (Table S1). Based on the needs of the research, the category in the land use data were divided into wetlands and non-wetlands. The types of wetlands in the study were ditch, river, pond, beach, paddy field, and brine pan. The rest of the LULC types were included in non-wetlands (Figure 2). Then, random precision evaluation points were created, we selected 2000 verify grids, and after field verification and the high-resolution remote sensing image test, the accuracy of the data interpretation for each year was more than 85%, which met the precision requirement of the study.

**Table 1.** Landsat images under in this study.

| Time | Precision (m) | Data Source | Cloudage (%) |
| --- | --- | --- | --- |
| 15 July 2009 | 30 | Landsat5 TM | 0.38 |
| 26 May 2014 | 30 | Landsat8 OLI_TIRS | 0.07 |
| 24 April 2020 | 30 | Landsat8 OLI_TIRS | 0.1 |

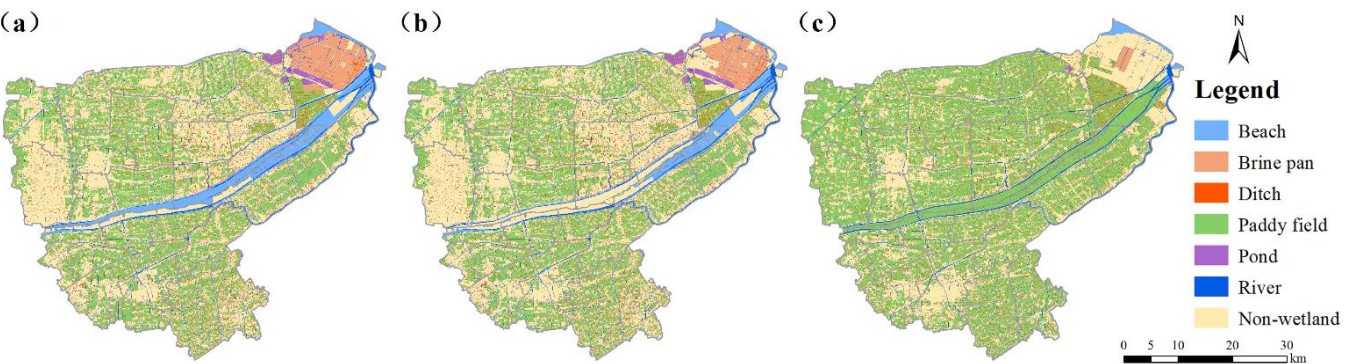

**Figure 2.** The present situation of LULC in different years: (**a**) 2009; (**b**) 2014; (**c**) 2020.

Based on the image data and land use status data, the boundaries of the city, village, port, industrial land, and coastline were extracted by ArcGIS 10.3. According to the present situation of LULC, the urban agglomeration areas were extracted as the city center, and the relevant distance data were obtained by calculating the distance between the patches of each wetland type and the geometric center of the city center, port, coastline, and industrial land.

The socio-economic data including the total population and gross domestic product (GDP) in 2009, 2014, and 2020 were obtained from the Jiangsu Statistical Yearbook, Lianyungang Yearbook, Guannan Yearbook, and Guanyun Yearbook. The obtained social and economic data of each town were transferred to the administrative region by using ArcGIS 10.3 and then rasterized so that each grid unit has corresponding data.

*2.3. Methods*

This paper studied the wetland landscape pattern change in the research area through the landscape dynamic rate, landscape conversion matrix, and landscape indices. At the same time, the index of human disturbance was used to analyze the change in human interference. Finally, the corresponding wetland landscape pattern to the human interference was quantitatively calculated by using the quadratic regression equation (Figure 3).

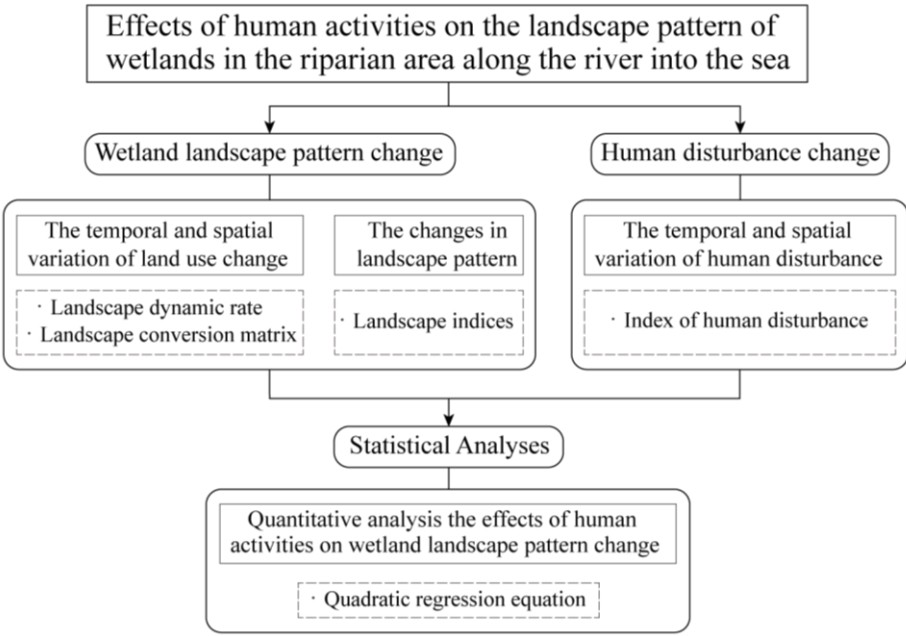

**Figure 3.** The technology roadmap of the study.

### 2.3.1. Landscape Dynamic Rate

The landscape dynamic rate describes the quantitative changes of a certain/integrated landscape type within a certain time range in the study area. The wetland landscape dynamic rate and integrated wetland landscape dynamic rate are important indicators to reflect the range and speed of wetland landscape pattern change. The formulas for both indicators are as follows [21]:

$$K = \frac{U_b - U_a}{U_a} \times \frac{1}{T} \times 100\%$$ (1)

$$LC = \left( \frac{\sum_{i=1}^{n} \Delta LU_{i-j}}{2\sum_{i=1}^{n} LU_i} \right) \times \frac{1}{T} \times 100\%$$ (2)

where $K$ is the dynamic rate of a certain wetland landscape type; $U_a$ and $U_b$ represent the area of a certain landscape type at the initial and last stages, respectively; $LC$ is the dynamic rate of the integrated wetland landscape; $LU_i$ is the area of the landscape at the initial stage; $\Delta LU_{i-j}$ is the absolute value of the area that landscapes convert into other types during the study period; $T$ is the duration of the research, and if $T$ represents time in years, $K$ and $LC$ denote the annual dynamic rate of a certain wetland landscape type and the integrated wetland landscape, respectively.

### 2.3.2. Landscape Conversion Matrix

The conversion matrix has been widely adopted to study LULC changes as it not only includes the area data of each category at a certain time in a certain region, but also has the information of the area transfer out of each category at the beginning of the period and the area transfer in each category at the end of the period. It reflects the dynamic process of the mutual conversion between all kinds of categories at the beginning and the end of a certain period in a certain region. Furthermore, the conversion matrix can fully and specifically

describe the structural characteristics of regional landscape change as well as the direction of conversion among all kinds of types. The conversion matrix is as follows [22,23]:

$$S_{ij} = \begin{bmatrix} S_{11} & S_{12} & \dots & S_{1n} \\ S_{21} & S_{22} & \dots & S_{2n} \\ \dots & \dots & \dots & \dots \\ S_{n1} & S_{n2} & \dots & S_{nn} \end{bmatrix} \tag{3}$$

where $S$ represents the area; $n$ is the number of landscape types before and after transfer; $i$ and $j$ ($i, j = 1, 2, \dots, n$) represent the landscape types before and after transformation, respectively; $S_{ij}$ represents the area that landscape $i$ converts to the landscape $j$. Each row of the element in the matrix represents the flow information of the $i$ landscape type before the transfer to the types after the transfer, and each column element represents the source information of the $j$ landscape type after the transfer from the types before the transfer.

2.3.3. Landscape Indices

The landscape index highly condenses the information on landscape patterns and can be used to quantitatively express the composition of landscape structure and spatial evolution. Furthermore, the landscape index reflects the landscape pattern concisely and is constructed from three levels: patch, landscape class, and landscape [34]. Because of redundancy between different landscape indices, some indices reflect similar landscape pattern information and some certain landscape indices are capable of representing the overall landscape pattern information. Hence, two principles are observed: (1) the inclusion of different types of indices and as much information as possible and (2) the avoidance of highly-correlated indices to avoid repeated calculation [35].

In addition, in order to better explore the evolution characteristics of regional landscape patterns, the study comprehensively considered the fragmentation, heterogeneity, complexity of the shape, aggregation, and dispersion, number of types, and balance of the distribution of landscape and selected five kinds of the index on the landscape class level: number of patches (NP), patch density (PD), edge density (ED), patch cohesion index (COHESION), and interspersion and juxtaposition index (IJI). On the landscape level, it removed the patch cohesion index (COHESION) and added another index: Shannon's diversity index (SHDI). The specific landscape index and meaning are shown in Table 2, and each index was calculated using FRAGSTAS 4.2 (http://www.umass. edu/landeco/research/fragstats/downloads/fragstats_downloads.html#FRAGSTATS, accessed on 15 January 2022).

**Table 2.** The selected landscape indices at the class and landscape levels.

| Category | METRIC | Description | Range | Scale |
|---|---|---|---|---|
| Landscape fragmentation indices | Number of patches (NP) | Equals the number of patches in the landscape or of the corresponding patch type. | NP ≥ 1, without limit. | C & L |
| | Patch density (PD) | The number of patches per 100 hectares. | PD > 0 | C & L |
| Landscape shape index | Edge density (ED) | Length of patches edge on a per unit area. It gets bigger when the landscape becomes more fragmental. | ED ≥ 0, without limit | C & L |
| Landscape convergence indices | Patch cohesion index (COHESION) | It measures the physical connectedness of the corresponding patch type. | 0 < COHESION ≤ 100 | C |
| | Interspersion and juxtaposition index (IJI) | It is based on patch adjacencies and isolates the interspersion or intermixing of patch types. | 0 < IJI ≤ 100 | C & L |
| Landscape diversity index | Shannon's diversity index (SHDI) | Equals minus the sum, across all patch types, of the proportional abundance of each patch type multiplied by that proportion. | SHDI ≥ 0, without limit | L |

C—Class level, L—Landscape level.

### 2.3.4. Index of Human Disturbance

Human disturbance generated by different LULC types and intensities has regional and cumulative characteristics. The analysis of human disturbance in the ecological environment based on the classification of land cover can comprehensively evaluate the cumulative results of various artificial effects and can accurately show the spatial distribution and gradient change characteristics of artificial influences. Based on previous research, the existing classification standards of human disturbance, and the actual situation of the study area, the corresponding hemeroby index (HI) was assigned to different types of LULC [24,36], and the assigned LULC types were divided into three categories: undisturbed, partially disturbed, and completely disturbed (Table 3).

**Table 3.** Hierarchy of the LULC type with respect to the hemeroby index (HI).

| Degree of Hemeroby | LULC Type | Hemeroby Index (HI) |
|---|---|---|
| Undisturbed (almost undisturbed by humans) | Other unused land | 0.14 |
| | Beach | 0.17 |
| | River | 0.23 |
| Partially disturbed (where human and nature impacts played equal roles such as crop or fishery ecosystems) | Pond | 0.30 |
| | Ditch | 0.50 |
| | Garden | 0.55 |
| | Woodland | 0.55 |
| | Grassland | 0.58 |
| | Paddy field | 0.65 |
| | Other arable land | 0.65 |
| | Dryland | 0.70 |
| | Brine pan | 0.75 |
| Completely disturbed (manmade entities like paved roads, etc.) | Traffic land | 0.95 |
| | Construction land | 0.99 |

Then, we used the Create Fishnet function in ArcGIS 10.3 to create grid units and a grid of 1 km × 1 km with a total of 2732 as the evaluation unit was constructed for sufficient sample size and accuracy. Next, according to Table 3, the disturbance index of different LULC types was assigned, and the human disturbance index of each grid unit can be calculated by Equation (4) [37].

$$A = \frac{\sum\limits_{i=1}^{n} HI_i \times D_i}{D} \tag{4}$$

where $A$ represents the human disturbance index of a single grid unit; $HI_i$ is the hemeroby index of the $i$ LULC type; $D_i$ is the area of the $i$ LULC type within the grid unit; $D$ is the total area of the grid unit; $n$ denotes the number of LULC types in the grid unit.

According to the human disturbance index of each grid unit acquired by the above method, the hemeroby stable index (HSI) in the study area from 2009 to 2020 was calculated. The formula is as follows:

$$S = \frac{\sum\limits_{i=1}^{m} |A_m - A_{m-1}|}{m} \tag{5}$$

where $S$ is the hemeroby stable index; $A_m$ and $A_{m-1}$ represent the human disturbance index of m year and $m-1$ year, respectively. The classification map of the hemeroby stable index in the study area was then generated according to the same method described above.

### 2.3.5. Statistical Analyses

The quadratic regression method can be used to study the nonlinear relationship and has been widely used in many fields such as biology, landscape, and economy [38–40], but it has not been used to study the relationship between the landscape pattern and human

disturbance. Therefore, this paper used this method to study the effects of human activities on the landscape pattern of wetlands in the riparian area along the river into the sea. In addition, this paper used panel data to analyze the relationship between landscape patterns and human disturbance, which has both section dimensions and time dimensions and can improve the accuracy of the estimation [41,42]. As far as this study is concerned, the cross-sectional data can tell us the landscape pattern index and human disturbance index of different grids at a certain time point, and the time series can tell us the landscape pattern index and human disturbance index of a certain grid in different years. When combined, we can obtain the change of landscape pattern and the human disturbance index at different times and samples.

To estimate the impacts of human activities on the wetland landscape pattern, the following fixed effect models were constructed:

$$y_{it} = \alpha X_{it} + \beta X_{it}^2 + \gamma Z_{it} + \mu_i + \varepsilon_{it} \tag{6}$$

where $i$ and $t$ represent different districts and different years, respectively. The dependent variable $y$ denotes the wetland landscape pattern including the number of patches (NP), patch density (PD), edge density (ED), interspersion and juxtaposition index (IJI), and Shannon's diversity index (SHDI). The independent variable $X$ represents the human disturbance index, and according to previous studies, it is speculated that the relationship between the landscape index and human disturbance index is not a simple linear relationship [43], so the quadratic term of the human disturbance index is also added. The control variable $Z$ contains the variables of construction land scale, cultivated land scale, total population, gross domestic product (GDP), distance to the city center, distance to the port, distance to the coastline, and distance to the industrial land. $\mu_i$ is the unit-specific error term, while $\varepsilon_{it}$ is the usual error term. Table 4 presents the definition of the independent and control variables and the dependent variables are described in Table 2. Descriptive statistics were used to analyze the data by using the Stata 15.1 software package (Stata Corporation, College Station, TX, USA).

**Table 4.** Definition of variables.

| Variable | | Definition | Units |
|---|---|---|---|
| Independent variables | Human disturbance index | It describes the impact index of human activities. | - |
| | Quadratic term of human disturbance index | It is the quadratic term for the human disturbance index. | - |
| Control variables | Construction land scale | Area of construction land. | km$^2$ |
| | Cultivated land scale | Area of farmland. | km$^2$ |
| | Total population | It is the sum of the population groups living in a certain area and a certain time range. | ten thousand people |
| | Gross domestic product (GDP) | It is the final result of the production activities of all permanent resident units in a country (or region) over a certain period. | hundred million RMB |
| | Distance to the city center | It is the distance from one place in the study area to the center of the city. | km |
| | Distance to the port | It is the distance from one place in the study area to the port. | km |
| | Distance to the coastline | It is the distance from one place in the study area to the coastline. | km |
| | Distance to the industrial land | It is the distance from one place in the study area to the industrial land. | km |

## 3. Results

### 3.1. Analysis of Landscape Pattern Change

#### 3.1.1. The Temporal and Spatial Changes in LULC

In 2009, 2014, and 2020, the total area of wetlands was 147,786.74 hm$^2$, 139,418.82 hm$^2$, and 177,295.93 hm$^2$, respectively. Additionally, in 2009 and 2014, the wetland types with a

relatively large area were paddy field, beach, and ditch, while in 2020, they were mainly paddy field, ditch, and river.

The results of the landscape pattern's change process in the study area from 2009 to 2020 are shown in Figure 4 and Table 5. From 2009 to 2014, only the area of non-wetland increased and the dynamic rates of other landscape types were in the order of beach (−4.65%) > brine pan (−4.08%) > pond (−0.29%) > paddy field (−0.18%) > river (−0.13%). From 2014 to 2020, the area of the paddy field showed a significant increasing trend with a dynamic rate of 9.43%, and the area of the ditch had the same trend, although the growth area was not so much, and the dynamic rate was 0.70%. Other landscape types decreased during these seven years and the dynamic rate of brine pan (−12.22%) was the highest, followed by beach (−12.05%) and then non-wetland (−4.65%).

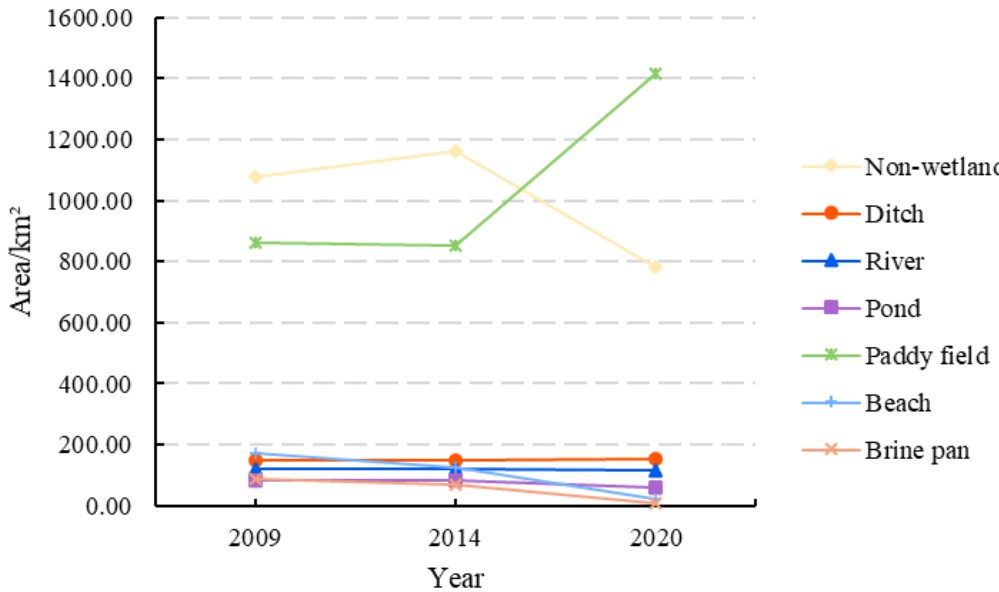

**Figure 4.** Changes in the area of different landscape types in the study area during 2009–2020.

**Table 5.** The dynamic rate of different landscape types in the study area during 2009–2020.

| Landscape Types | 2009–2014 | 2014–2020 | 2009–2020 |
|---|---|---|---|
| Non-wetland | 1.29% | −4.65% | −2.28% |
| Ditch | −0.24% | 0.70% | 0.28% |
| River | −0.13% | −0.30% | −0.24% |
| Pond | −0.29% | −4.51% | −2.73% |
| Paddy field | −0.18% | 9.43% | 5.35% |
| Beach | −4.65% | −12.05% | −7.39% |
| Brine pan | −4.08% | −12.22% | −7.42% |

During the 2009–2020 period, the area of non-wetland increased from 2009 to 2014 and decreased from 2014 to 2020. In contrast, the area of paddy fields and ditch decreased from 2009 to 2014 and increased from 2014 to 2020. Other landscape types were reduced between 2009 and 2020. Overall, from 2009 to 2020, only the areas of paddy fields and ditch increased, with dynamic rates of 5.35% and 0.28%, respectively, and the area of the other landscape types including non-wetland, river, pond, beach, and brine pan decreased, with dynamic rates of −2.28%, −0.24%, −2.73%, −7.39%, and −7.42%, respectively.

At the same time, during the 2009–2014 and 2014–2020 periods, the integrated landscape dynamic rates were 0.55% and 3.18%, respectively. It showed an increasing trend and indicated that the change in LULC was faster between 2014 and 2020.

The spatial change trend is shown in Figure 5, where the change in the landscape pattern was relatively stable and mainly occurred in the central and northeast of the study

area from 2009 to 2014 (Figure 5a). Table 6 shows that the major wetlands changed into non-wetlands during the 2009–2014 period, special beach (4780.38 hm$^2$), brine pan (2188.97 hm$^2$), and paddy field (1022.01 hm$^2$). While the total conversion area of non-wetland to wetland was really small, among which the paddy field was the largest, with a conversion area of 54.25 hm$^2$. From 2014 to 2020, the landscape pattern transformation was dramatic and mainly occurred in the northwest, northeast, and center of the study area (Figure 5b). From Table 7, the conversion area of wetland to non-wetland was larger than the area of non-wetland to wetland, and the degraded non-wetland was mostly converted into a paddy field, where the conversion area was 56,649.14 hm$^2$. The areas of the wetland to non-wetland in the study area were in the order of paddy field (8589.95 hm$^2$) > brine pan (5670.68 hm$^2$) > pond (3367.54 hm$^2$) > beach (2981.33 hm$^2$) > ditch (2174.03 hm$^2$) > river (848.34 hm$^2$).

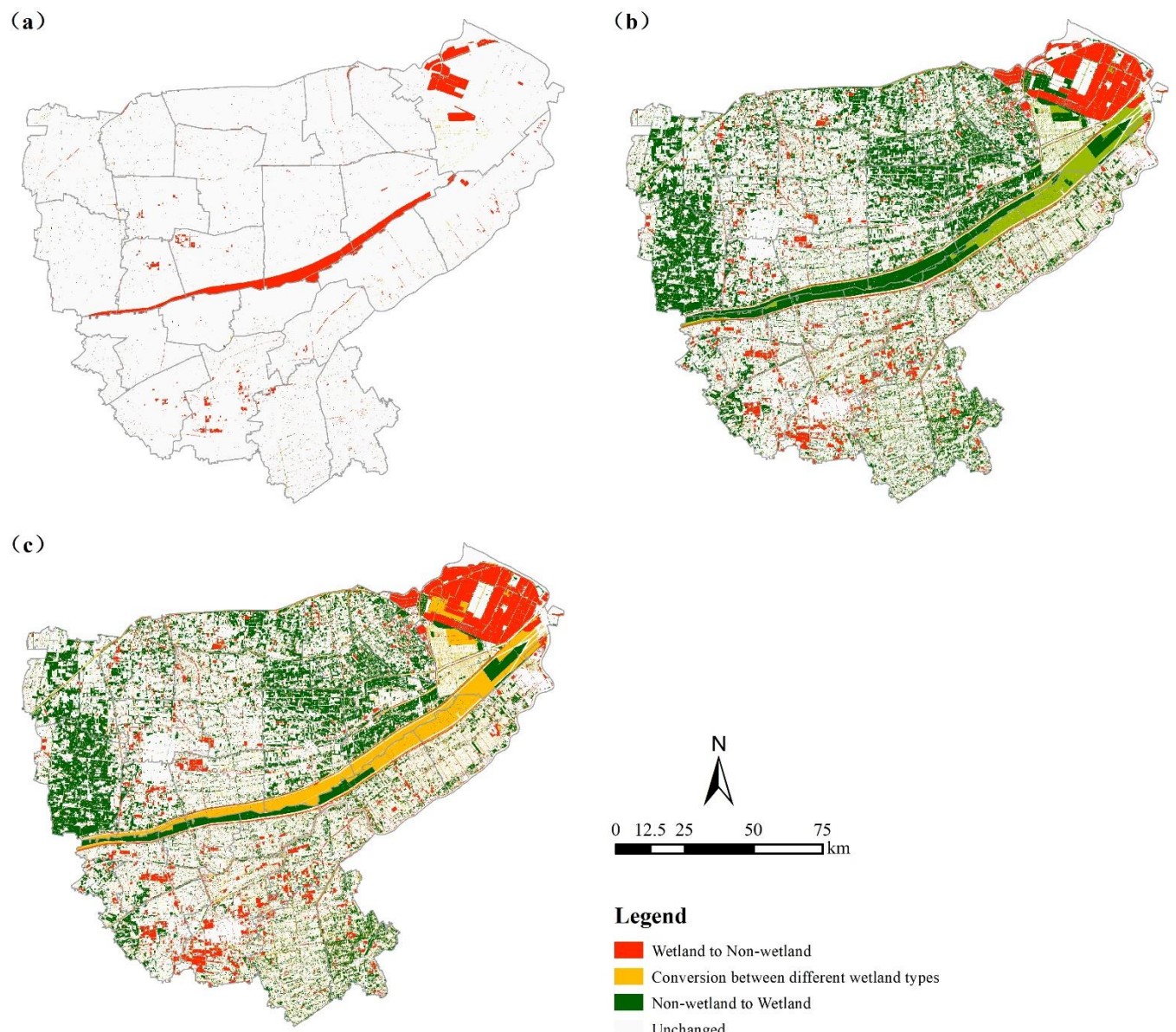

**Figure 5.** Landscape conversion matrix of different years: (**a**) 2009–2014; (**b**) 2014–2020; (**c**) 2009–2020.

**Table 6.** Area of the landscape conversion matrix from 2009 to 2014 (hm$^2$).

| 2009 | 2014 | | | | | | | |
|---|---|---|---|---|---|---|---|---|
| | Non-Wetland | Ditch | River | Pond | Paddy Field | Beach | Brine Pan | Total Area |
| Non-wetland | 107,834.30 | 1.15 | 2.28 | 0.47 | 54.25 | 0.05 | 3.13 | 107,895.63 |
| Ditch | 198.92 | 14,775.25 | 0.01 | 0.04 | 19.01 | 0.01 | 0.01 | 14,993.25 |
| River | 93.07 | 0.01 | 11,943.05 | 0.00 | 0.92 | 0.05 | 0.01 | 12,037.11 |
| Pond | 145.90 | 0.04 | 0.01 | 8322.27 | 4.27 | 0.00 | 0.00 | 8472.49 |
| Paddy field | 1022.01 | 1.01 | 0.06 | 0.09 | 85,137.04 | 0.01 | 0.00 | 86,160.22 |
| Beach | 4780.38 | 0.01 | 0.20 | 0.00 | 15.29 | 12,408.50 | 0.00 | 17,204.38 |
| Brine pan | 2188.97 | 0.01 | 0.01 | 0.01 | 0.00 | 0.00 | 6730.29 | 8919.29 |
| Total area | 116,263.55 | 14,777.48 | 11,945.62 | 8322.88 | 85,230.78 | 12,408.62 | 6733.44 | 255,682.37 |

**Table 7.** Area of landscape conversion matrix from 2014 to 2020 (hm$^2$).

| 2014 | 2020 | | | | | | | |
|---|---|---|---|---|---|---|---|---|
| | Non-Wetland | Ditch | River | Pond | Paddy Field | Beach | Brine Pan | Total Area |
| Non-wetland | 54,754.57 | 3085.26 | 570.33 | 1062.02 | 56,649.14 | 78.86 | 63.37 | 116,263.55 |
| Ditch | 2174.03 | 9842.61 | 48.60 | 665.80 | 2044.07 | 2.19 | 0.18 | 14,777.48 |
| River | 848.34 | 217.68 | 10,282.36 | 91.93 | 420.46 | 72.83 | 12.02 | 11,945.62 |
| Pond | 3367.54 | 545.74 | 22.93 | 3377.53 | 998.85 | 10.29 | 0.00 | 8322.88 |
| Paddy field | 8589.95 | 1606.40 | 37.65 | 238.84 | 74,757.76 | 0.18 | 0.00 | 85,230.78 |
| Beach | 2981.33 | 176.05 | 692.78 | 181.56 | 6603.62 | 1773.28 | 0.00 | 12,408.62 |
| Brine pan | 5670.68 | 27.26 | 41.04 | 78.88 | 18.05 | 0.00 | 897.53 | 6733.44 |
| Total area | 78,386.44 | 15,501.00 | 11,695.69 | 5696.56 | 141,491.95 | 1937.63 | 973.10 | 255,682.37 |

The landscape conversion occurred mostly in the northwest, northeast, and middle of the study area from 2009 to 2020, and the northern part of the study area was mainly non-wetland to wetland, while the southern part was just the opposite (Figure 5c). Among all of the landscape types, the paddy field was the landscape type with the largest conversion area, whether transformed from non-wetland (51,348.64 hm$^2$) or converted to non-wetland (9471.83 hm$^2$), and the conversion areas of other landscape types showed an increasing trend during the study period (Table 8).

**Table 8.** Area of the landscape conversion matrix from 2009 to 2020 (hm$^2$).

| 2009 | 2020 | | | | | | | |
|---|---|---|---|---|---|---|---|---|
| | Non-Wetland | Ditch | River | Pond | Paddy Field | Beach | Brine Pan | Total Area |
| Non-wetland | 52,256.46 | 2665.16 | 514.37 | 1024.46 | 51,348.64 | 58.42 | 28.12 | 107,895.63 |
| Ditch | 2319.69 | 9869.25 | 49.03 | 669.52 | 2083.29 | 2.29 | 0.18 | 14,993.25 |
| River | 909.65 | 231.67 | 10,292.38 | 91.75 | 426.77 | 72.87 | 12.02 | 12,037.11 |
| Pond | 3457.88 | 550.58 | 23.37 | 3383.53 | 1046.84 | 10.29 | 0.00 | 8472.49 |
| Paddy field | 9471.83 | 1619.32 | 38.23 | 250.43 | 74,780.23 | 0.18 | 0.00 | 86,160.22 |
| Beach | 3135.16 | 345.44 | 730.21 | 184.01 | 11,033.39 | 1776.17 | 0.00 | 17,204.38 |
| Brine pan | 6835.77 | 219.58 | 48.10 | 92.86 | 772.79 | 17.41 | 932.78 | 8919.29 |
| Total area | 78,386.44 | 15,501.00 | 11,695.69 | 5696.56 | 141,491.95 | 1937.63 | 973.10 | 255,682.37 |

### 3.1.2. The Variation of Landscape Pattern

From the perspective of landscape (Table 9), the landscape pattern of the study area has changed significantly due to the impact of human activities. During the 2009–2020 period, the values of NP and PD showed a dynamic process of decreasing–increasing, indicating that the fragmentation of the landscape pattern reduced from 2009 to 2014 and then intensified from 2014 to 2020. The value of ED also showed the same trend as NP and PD, indicating that the complexity of the boundary shape of the overall landscape decreased from 2009 to 2014; after 2014, the landscape shape tended to develop irregularly.

The IJI index showed an overall downward trend, and its value dropped rapidly from 2014 to 2020, indicating that the connectivity of patches in the landscape decreased and all types of patches were scattered. The value of SHDI also decreased from 2009 to 2020, indicating that the landscape diversity and uniformity of the area occupied by various types of patches decreased between 2009 and 2020.

**Table 9.** The dynamic variation of landscape index in the study area in 2009–2020.

| Year | NP | PD | ED | IJI | SHDI |
|------|------|--------|--------|---------|--------|
| 2009 | 51,744 | 11.7028 | 17.7604 | 54.1306 | 1.4566 |
| 2014 | 51,346 | 11.6128 | 16.8157 | 52.2593 | 1.4201 |
| 2020 | 51,426 | 11.6309 | 24.3261 | 35.5374 | 1.1367 |

From the perspective of the landscape class type, the values of NP and PD had the same trend and are shown in Figure 6a,b. The NP and PD of the ditch and pond were larger, in which ditch was the largest one and was the only landscape type that grew in the 2014–2020 period, indicating that the fragmentation of ditch and pond was serious. Although the NP and PD of the landscape types other than ditch fluctuated from 2009 to 2020, it showed a decreasing trend in general.

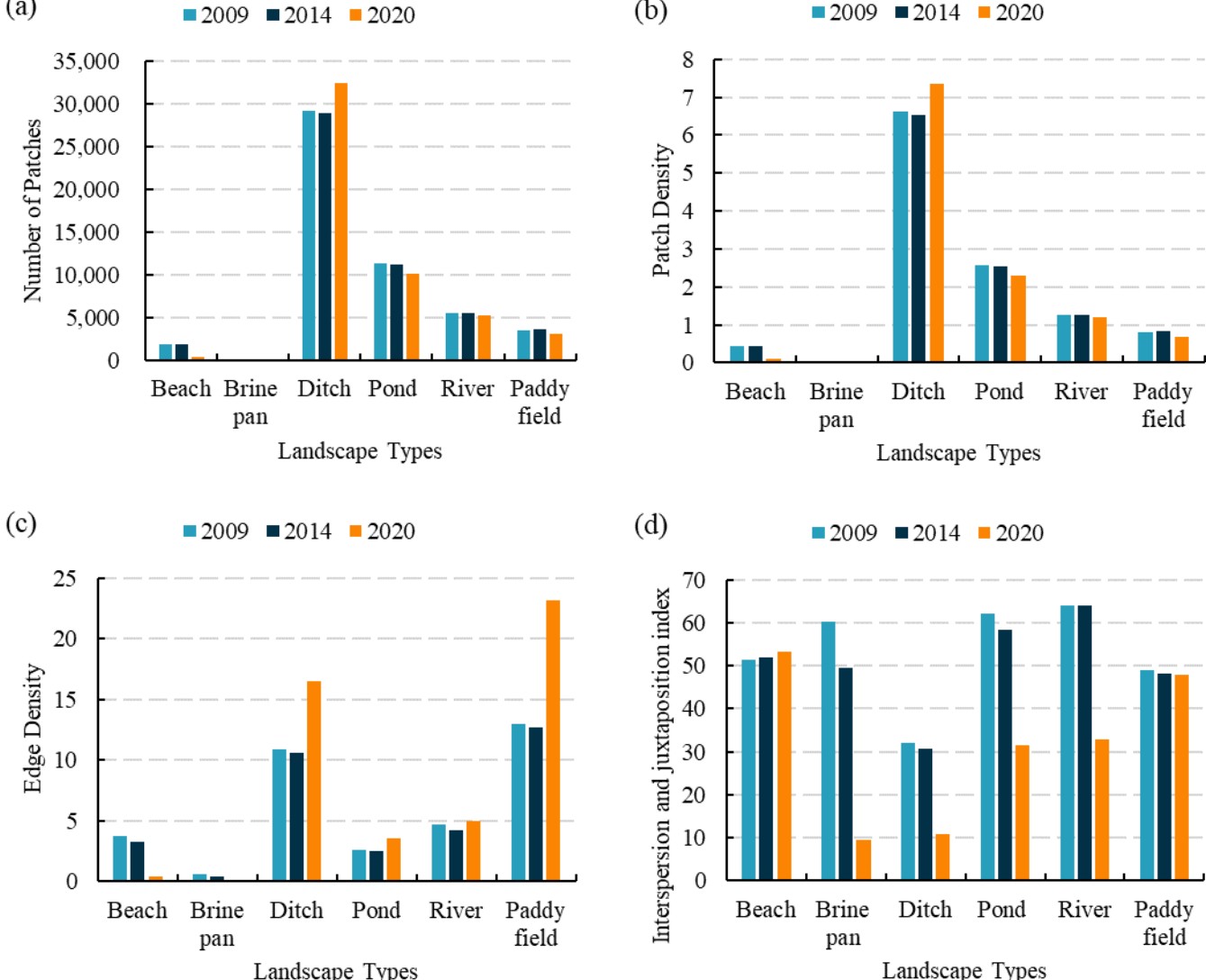

**Figure 6.** *Cont.*

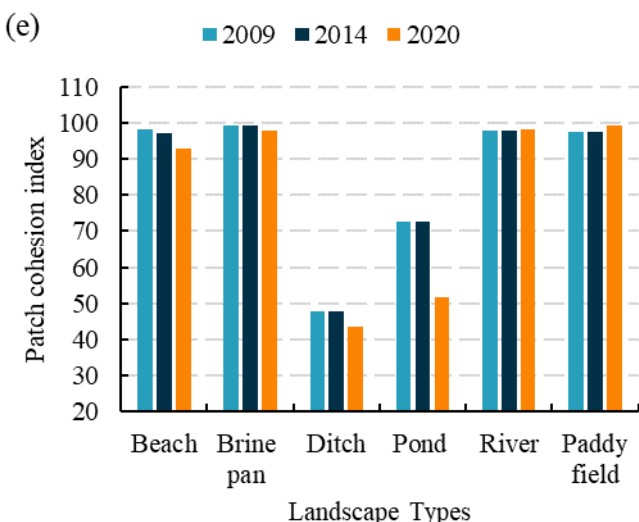

**Figure 6.** Landscape indices on the landscape class level in the study area from 2009 to 2020: (**a**) change in NP; (**b**) change in PD; (**c**) change in ED; (**d**) change in IJI; (**e**) change in COHESION.

As shown in Figure 6c, the landscape types that had larger ED in the study area were paddy fields, ditch, rivers, and ponds. It is indicated that the edges of paddy fields, ditch, rivers, and ponds were more complex and affected by human activities. In addition, the ED of those four landscape types decreased from 2009 to 2014 and increased from 2014 to 2020, which indicated that the landscape shape of the four landscape types became a bit simpler in the 2009–2014 period and then become more complicated in the 2014–2020 period, whereas the ED of the beach and brine pan decreased from 2009 to 2020, the decline in ED meant that the total perimeter of the patches became smaller, and the patch shape developed regularly and was simplified.

The changes in IJI and COHESION for the study area are shown in Figure 6d,e. IJI is the interspersion and juxtaposition between one certain patch type and other patch types at the landscape class level. From 2009 to 2014, the IJI of all of the landscape types except the ditch was large, indicating that the distribution of these landscape types was more gathered and adjacent to other types. In the stage of 2014–2020, only the IJI of the beaches increased, while others showed a downward trend and their distributions were more discrete, with a few adjacent to other types. In short, the landscape of the study became more dispersive among different landscape types. COHESION is the cohesion of the same landscape type at the landscape class level. The COHESION of beach, brine pan, river, and paddy field was expressed much larger, indicating that the connectivity of these landscape types was at a high level. In contrast, the COHESION of the ditch and ponds was a bit small and showed a rapid decline from 2014 to 2020, which indicated that these landscape types were more scattered, and the connectivity between certain landscape types slowed down. In a word, except for a slight increase in the landscape connectivity in the river and paddy fields, the connectivity and aggregation decreased among the patches with the same landscape type from 2009 to 2020.

### 3.2. Analysis of Temporal and Spatial Variation of Human Disturbance

3.2.1. The Changes of Human Disturbance in the Temporal Dimension

According to the method above-mentioned, the human disturbance indices of 2732 statistical grids (1 km × 1 km) were calculated in ArcGIS 10.3 software, showing that the mean human disturbance indices of the whole study area were 0.6075, 0.6186, and 0.6123, respectively, in the three target years (Figure 7).

From 2009 to 2014, the human disturbance index showed a significant upward trend, and after 2014, the human disturbance index began to decline slowly. This indicates that the intensity of human disturbance in the whole study area increased in the 2009–2014 period

and decreased in the 2014–2020 period. During the 2009–2020 period, the landscape types of different degrees of hemeroby changed significantly, which mainly showed that the area of undisturbed elements decreased year by year, while the area of partially disturbed elements increased gradually, and the area of completely disturbed elements showed a dynamic process of increasing–decreasing. Overall, the area of partially disturbing elements from 2009 to 2020 had an absolute advantage, and the change here was most intense.

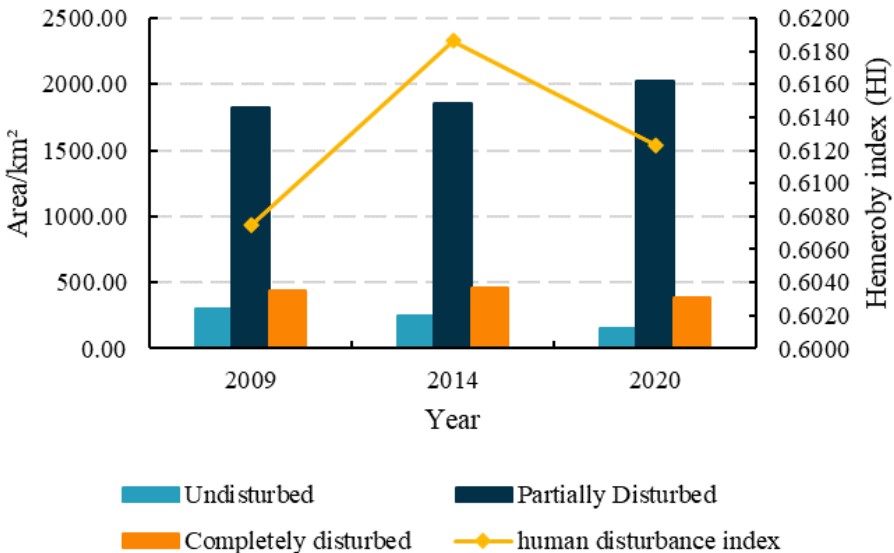

**Figure 7.** The human disturbance index of the study area from 2009 to 2020.

### 3.2.2. The Changes of Human Disturbance in the Spatial Dimension

The high interference areas were mainly distributed in the northwest, southwest, and northeast of the study area in 2009, and there were five high interference nuclei in these areas, which are mostly urban centers and settlements. The low interference areas are mainly distributed in the middle of the study area, mostly along the river (Figure 8a). In 2014, the area of high interference increased slightly and expanded from those above-mentioned five high interference nuclei in the study area. In addition, the human disturbance index of the area along the river in the middle of the study area also increased (Figure 8b). In 2020, the human disturbance index of the two city centers in the northwest and southwest was still relatively high, while the other high interference nuclei reduced or even disappeared, and the areas with high interference became more scattered (Figure 8c). On the whole, the areas of high human disturbance were mainly concentrated in the places where towns and construction land gathered and formed a decreasing trend from the center to the outside.

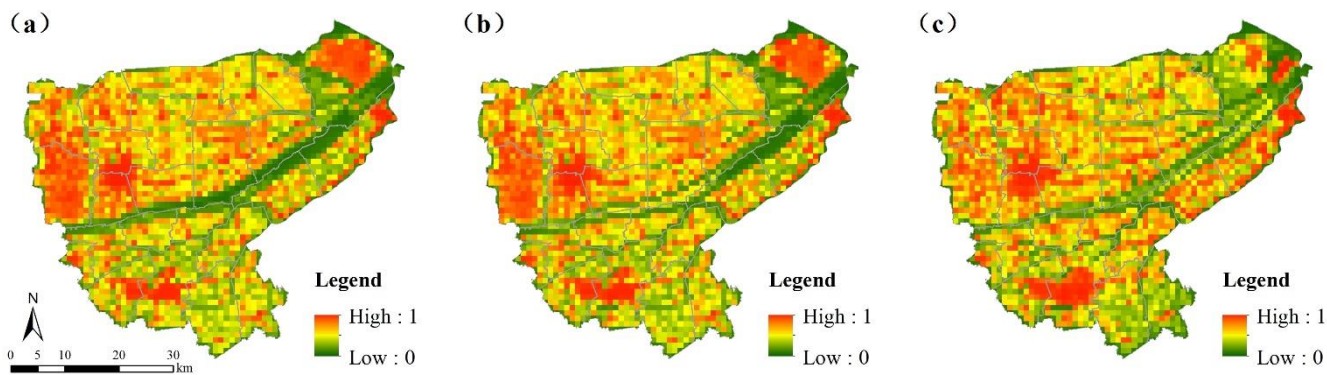

**Figure 8.** The spatial distribution of standardized human disturbance in the study area for three years: (**a**) 2009; (**b**) 2014; (**c**) 2020.

### 3.3. Regression Results of Landscape Index and Human Disturbance Index

The number of observations, mean value, standard deviation, minimum value, and maximum value of all dependent variables, independent variables, and control variables are shown in Table 10. The results of the impact of the human disturbance index on various landscape indices are presented in Table 11. The last line of Table 11 showed all the F-tests were statistically significant at the 1% level of significance, indicating the overall models were significant. This also shows that the quadratic regression equation is effective in studying the influencing factors of wetland landscape patterns. Column (1) showed that the coefficient of the human disturbance index was 55.921, while the coefficient of the quadratic term of the human disturbance index was −57.292, and both of them were statistically significant at the 1% level of significance. The results indicated that with the increase in human interference, the NP showed a trend of first increasing and then decreasing. Similarly, columns (2)–(5) showed that the human disturbance index had a nonlinear impact on PD, ED, IJI, and SHDI. Additionally, the control variables had differentiated effects on different kinds of landscape indices. The total population had a significant negative effect on ED and a positive effect on SHDI. The GDP had a negative impact on NP and PD as well as SHDI. The NP, PD, ED, and IJI were positively correlated with the cultivated land scale and construction land scale, while SHDI was negatively correlated with the cultivated land scale and construction land scale. The distance to the city center showed significant positive correlations with ED, while it had negative correlations with SHDI. The distance to the port showed significant positive correlations with IJI, but negative correlations with ED and SHDI. The distance to the coastline exerted a positive impact on NP and PD. The distance to the industrial land had a positive influence on ED and IJI, while it had a negative influence on SHDI.

**Table 10.** Descriptive statistics of the variables.

| Variable | Observations | Mean | Std. Dev. | Min | Max |
|---|---|---|---|---|---|
| Human disturbance index | 8196 | 0.613 | 0.176 | 0.000 | 0.988 |
| Quadratic term of human disturbance index | 8196 | 0.406 | 0.519 | 0.000 | 0.976 |
| Total population | 8196 | 7.389 | 4.044 | 0.632 | 20.810 |
| Gross domestic product (GDP) | 8196 | 34.630 | 81.330 | 2.800 | 483.000 |
| Cultivated land scale | 8196 | 0.555 | 0.263 | 0.000 | 1.000 |
| Construction land scale | 8196 | 0.157 | 0.184 | 0.000 | 1.000 |
| Distance to the city center | 8196 | 17.880 | 13.860 | 0.006 | 57.010 |
| Distance to the port | 8196 | 9.046 | 5.193 | 0.087 | 26.950 |
| Distance to the coastline | 8196 | 41.380 | 18.730 | 0.218 | 71.800 |
| Distance to the industrial land | 8196 | 2.568 | 2.076 | 0.001 | 14.480 |
| NP | 8196 | 22.660 | 12.490 | 0.000 | 73.000 |
| PD | 8196 | 22.660 | 12.490 | 0.000 | 73.000 |
| ED | 8196 | 30.200 | 21.860 | 0.000 | 135.000 |
| IJI | 8196 | 41.790 | 24.510 | 0.000 | 100.000 |
| SHDI | 8196 | 0.699 | 0.333 | 0.000 | 1.565 |

**Table 11.** Regression results of the influence factors on the landscape index.

| Variables | (1) NP | (2) PD | (3) ED | (4) IJI | (5) SHDI |
|---|---|---|---|---|---|
| Human disturbance index | 55.921 *** (5.895) | 55.976 *** (5.896) | 40.769 ** (20.659) | 184.866 *** (32.641) | 3.732 *** (0.326) |
| Quadratic term of human disturbance index | −57.292 *** (5.949) | −57.348 *** (5.950) | −178.208 *** (20.962) | −203.271 *** (30.427) | −2.509 *** (0.300) |
| Total population | 0.039 (0.098) | 0.039 (0.098) | −4.058 *** (0.252) | −0.291 (0.373) | 0.051 *** (0.004) |

**Table 11.** *Cont.*

| Variables | (1) NP | (2) PD | (3) ED | (4) IJI | (5) SHDI |
|---|---|---|---|---|---|
| Gross domestic product (GDP) | −0.001 * | −0.001 ** | 0.001 | 0.001 | −0.000 *** |
| | (0.001) | (0.001) | (0.002) | (0.003) | (0.000) |
| Cultivated land scale | 2.470 *** | 2.468 *** | 52.231 *** | 9.883 ** | −0.579 *** |
| | (0.929) | (0.929) | (2.729) | (4.831) | (0.051) |
| Construction land scale | 3.914 *** | 3.923 *** | 62.755 *** | 39.152 *** | −0.481 *** |
| | (1.103) | (1.103) | (3.516) | (6.237) | (0.067) |
| Distance to the city center | −0.008 | −0.008 | 1.778 *** | 0.044 | −0.023 *** |
| | (0.070) | (0.070) | (0.188) | (0.313) | (0.004) |
| Distance to the port | −0.000 | −0.000 | −0.181 *** | 0.424 *** | −0.002 ** |
| | (0.014) | (0.014) | (0.042) | (0.074) | (0.001) |
| Distance to the coastline | 11.173 * | 11.162 * | −23.724 | −22.536 | 0.304 |
| | (6.178) | (6.178) | (18.589) | (21.279) | (0.205) |
| Distance to the industrial land | 0.046 | 0.046 | 0.317 *** | 0.653 *** | −0.004 ** |
| | (0.036) | (0.036) | (0.117) | (0.178) | (0.002) |
| Constant | −452.620 * | −452.193 * | 1015.251 | 926.241 | −12.613 |
| | (255.715) | (255.713) | (769.038) | (879.881) | (8.470) |
| Year FE | yes | yes | yes | yes | yes |
| Observations | 8196.000 | 8196.000 | 8196.000 | 8196.000 | 8196.000 |
| $R^2$ | 0.063 | 0.063 | 0.493 | 0.077 | 0.395 |
| F | 23.716 *** | 23.722 *** | 253.152 *** | 14.363 *** | 162.254 *** |

Notes: Standard errors in parentheses, * $p < 0.1$, ** $p < 0.05$, *** $p < 0.01$.

## 4. Discussion

*4.1. Driving Factors of the Spatial and Temporal Changes of the Wetland Landscape Pattern in the Riparian Area along the River into the Sea*

Wetland landscape evolution in the riparian zone of coastal regions is driven by both natural and human factors [44], so this paper mainly studied the impact of human activities on wetlands in these areas. It was found in the present study that the LULC in the study area changed spatially and temporally during the past 10 years. The change occurred mainly in the northwest, northeast, and middle of the study area, and the area of most wetland types continued to decrease, except for the ditch and paddy fields. In addition, the landscape indices changed differently, with the NP, PD, IJI, and SHDI, except for ED, decreasing between 2009 and 2020. These changes in the index indicate that the fragmentation, connectivity, and diversity of the wetland landscape patterns showed a downward trend on the whole.

For the driving forces from human activities, the effects were likely more drastic in a short time [45–47]. This study revealed several pathways of the driving effects of human activities on the wetland landscape evolution in coastal regions. First, the social and economic development in coastal regions is better than that in inner areas in general, especially in China [48]. The demand for construction lands such as residential land, industrial land, and land for public facilities is also increasing, which may increase the possibility of occupying the wetland space. As a result, the fragmentation of the landscape pattern was reduced and the complexity of the boundary shape of the overall landscape showed the same trend. Second, with the intensification of urban construction and the development and utilization of wetlands, the connectivity of the wetland landscape pattern was reduced, which could weaken the ecosystem function of wetlands. Third, as the location is near the river and sea, the riparian zone of coastal regions has the endowment advantage of agricultural and fishery production [49]. Furthermore, there are many paddy fields and ponds in the study area, which provide good conditions for agriculture and aquaculture. These activities could also reduce the diversity of the wetland landscape.

Additionally, with the continuous improvement in the residents' living standards, the demand for better environmental quality is also increasing. More attention is being paid

to the landscape function and ecosystem service of wetlands. As the green development concept and national ecological protection red line plan have been conducted across China, the coastal wetlands were given further protection. Since the wetlands are widely distributed in coastal regions, some wetland-parks have been developed in the area where the scenery is beautiful, which has a positive effect on the wetland landscape protection. These ecological protection measures are conducive to wetland protection, ecological restoration, and sustainable development. The driving forces of various factors on wetland landscape evolution in the riparian zone of a coastal region are summarized in Figure 9.

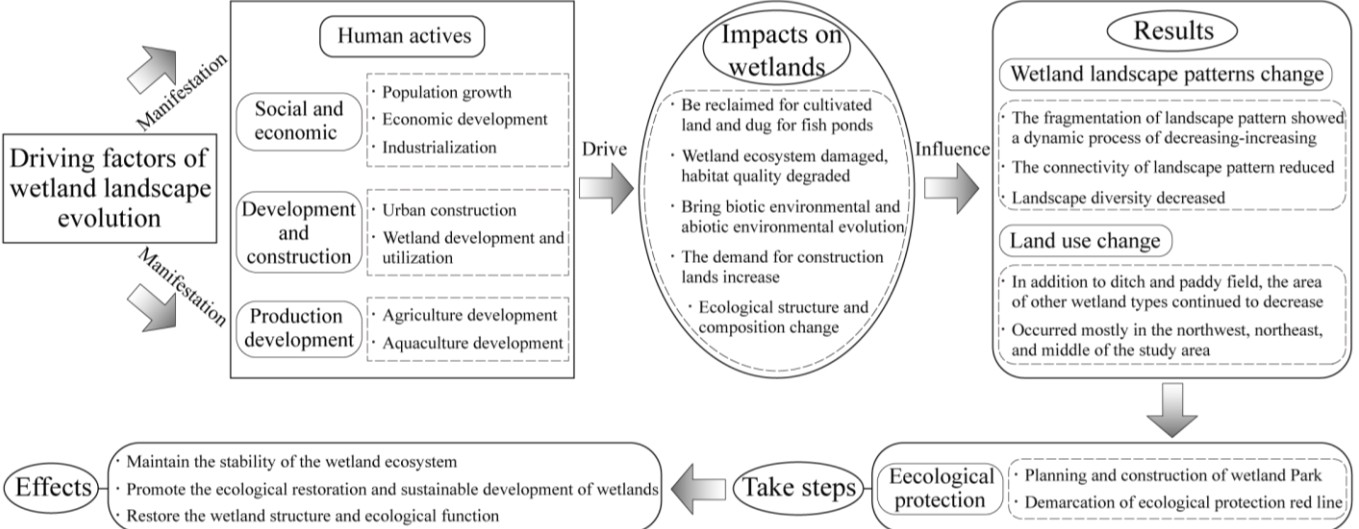

**Figure 9.** Driving factors and results of the evolution of the wetland landscape.

### 4.2. Changes in Human Activities on the Wetland Landscape Pattern in the Riparian Area along the River into the Sea

The results of the present study demonstrated that human disturbance changes have stage characteristics in temporal and spatial heterogeneous characteristics. In the period from 2009 to 2014, the human disturbance that mainly increased occurred near the rivers and towns, but decreased in small patches widely distributed in the study area. In the period from 2014 to 2020, the human disturbance was more drastic, emerging in the downstream runoff and near towns, but the decrease in the human disturbance was obvious locally, which could be seen in the west and northeast (Figure 10). As was found in the regression analysis, with the human disturbance increasing, the NP, PD, ED, IJI, and SHDI increased first and then decreased (Figure 11). The human disturbance index was consistent with the results obtained by adding all of the above-mentioned variables, and the models were robust. Therefore, for the sake of simplicity, the graph only shows the relationship between the human disturbance index, a key variable, and each landscape index. Additionally, human activities such as economic growth, population agglomeration, and rural and urban development could increase the wetlands' fragmentation degree and make shape of the wetlands more complex. According to the results, it has also been found that the wetlands near the city center and industrial land have higher variousness and a more regular shape. The increase in cultivated land area, especially the paddy field area, could help in the protection of wetlands for an improvement in the density and connectivity. It seems that in the initial phase, the human disturbances came from the destruction of the wetlands for development and construction, but human disturbances have recently appeared in both destruction and remediation.

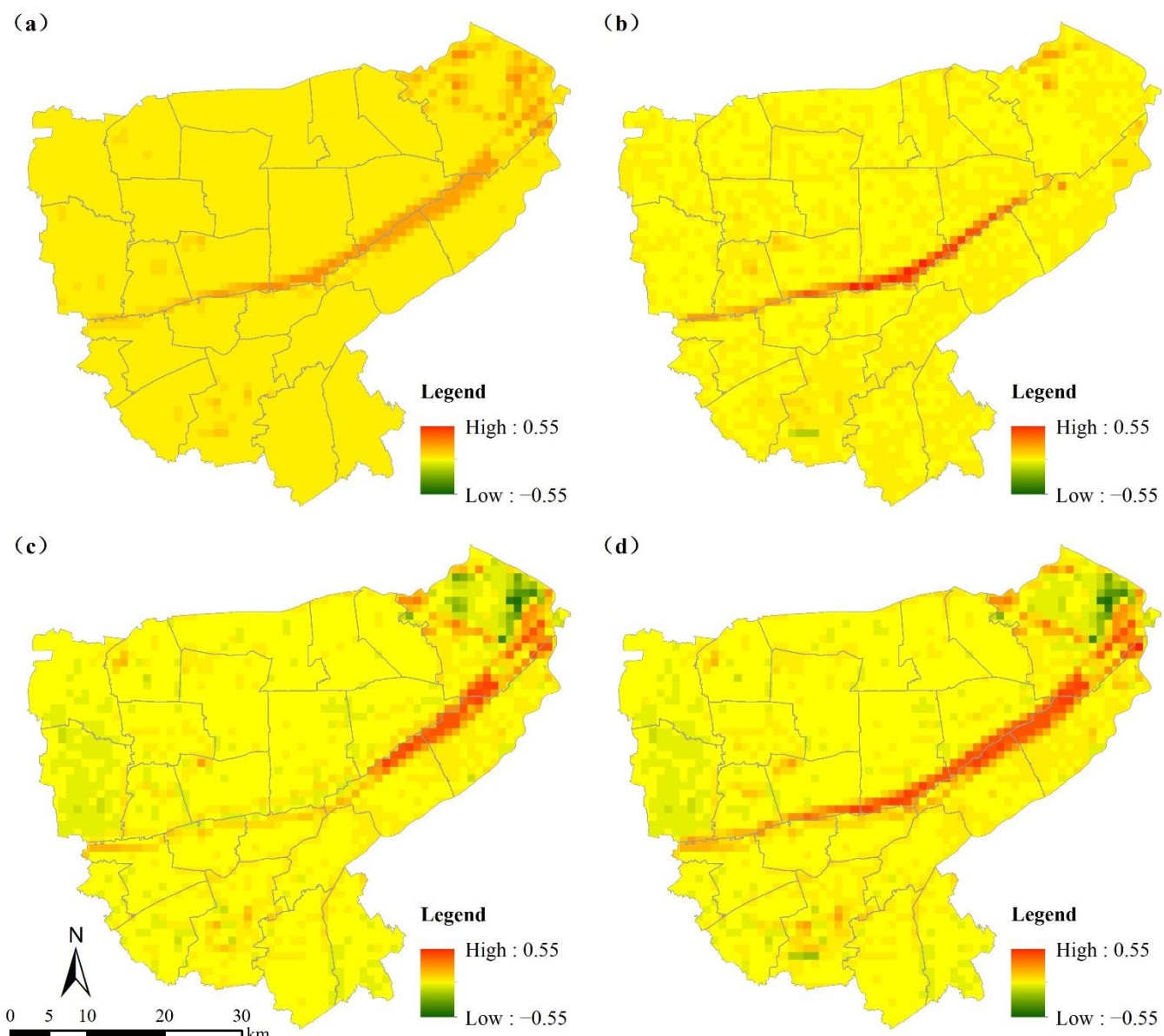

**Figure 10.** The changes in human disturbance in the spatial dimension from 2009 to 2020. (**a**) The human disturbance stability from 2009 to 2020. (**b**) The changes in the period from 2009 to 2014. (**c**) The changes in the period from 2014 to 2020; (**d**) The changes in the period from 2009 to 2020.

The wetland ecosystem has been improved in some areas of the region. The results stand in line with previous studies [43]. Furthermore, some researchers have used polynomial fitting, correlation analysis, and other methods to study the relationship between the human disturbance index and landscape indices, and obtained the nonlinear relationship, which provides the basis for the development of this study [20,50]. According to the characteristics of the inverted U-type relationships obtained by the quadratic regression equation, the measures of space control and ecological remediation should be taken in the region to keep the wetlands' density, connectivity, and diversity at a higher level. It is helpful to maintain sustainable development in the riparian area along the river in coastal regions.

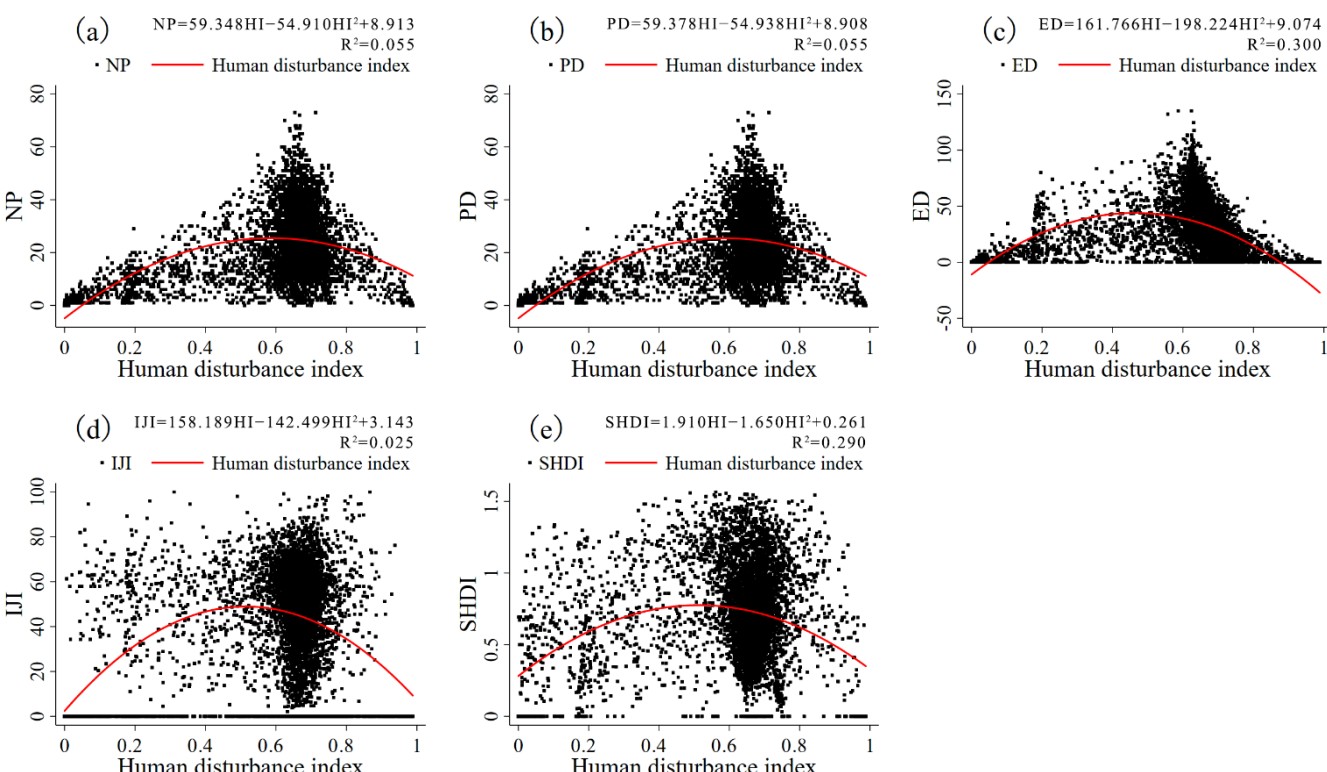

**Figure 11.** Relationship between the different landscape indices ((**a**) NP, (**b**) PD, (**c**) ED, (**d**) IJI, and (**e**) SHDI) and human disturbance index based on the landscape level.

### 4.3. Protections of the Wetland in the Riparian Zone of Coastal Regions

During the development and utilization of territorial space in the riparian zone of coastal regions, human disturbance and even the destruction of wetlands were unavoidable, as the wetlands were widely distributed and were an effective source of development space. How to take measures to protect the wetlands in this region to continuously provide ecosystem services according to the effects of human activities on wetland landscape evolution is critical for ecological improvement [51]. Additionally, the regression results conducted in the region also found that the cultivated land scale, construction land scale, distance to the city center, distance to the port, and distance to the industrial land affected the landscape change of the wetlands. Attention should be paid to the protection of wetlands in the process of land use in this area, and ecological remediation should be carried out to improve the structure and ecological function of the wetlands near the city center, ports, and industrial land.

In the case of the wetlands themselves, further enhanced protection and increased connectivity are needed, which is conducive to restoring the ecosystem function of the wetlands in the riparian area along the river into the sea. In addition, the study area is densely covered by the river network, and the supply of water is very important for wetlands, so a clean and stable water supply must be ensured to maintain the operation of wetland ecosystems. At the same time, the diversity of wetland landscape pattern types also needs to be given attention. Protecting the ecological environment structure and establishing buffer zones can be used to preserve wetlands to enrich the diversity of wetland landscape pattern types [25,52].

From the perspective of management, it is necessary to formulate relevant wetland protection policies and urban development and construction plans in advance to prevent wetland damage caused by the disorderly expansion of the city [51,53]. What is more important is to improve the sustainability of relevant plans and policies, and conduct regular assessments to ensure the later implementation. Coastal human activities including

reclamations should also be scientifically managed for incorporation with national or regional protection and development planning in the future. For areas where wetlands have been damaged, relevant procedures for the restoration of impaired coastal wetlands and ecological compensation need to be established to restore the ecosystem service of these wetlands. Furthermore, with the rapid development of the social economy, coordinating socio-economic development and environmental conservation is also an important part of protecting wetlands in the riparian area along the river into the sea.

## 5. Conclusions

In this study, we used the landscape dynamic rate, landscape conversion matrix, landscape indices, and human disturbance index to investigate the evolution of processes and the driving forces of landscape and human disturbance changes with all of the available Landsat images and socio-economic data in the study area. In addition, the quadratic regression equation was used to analyze the impact of human disturbance on the landscape pattern.

The results revealed that wetlands experienced drastic changes in the study area from 2009 to 2020. The area of the wetlands and the landscape fragmentation decreased during the 2009–2014 period and then increased during the 2014–2020 period. At the same time, the diversity and connectivity of the landscape decreased during the study period. For human activities, the human disturbance index showed a dynamic process of increasing–decreasing, and there were cores with high human interference in the northwest, southwest, and northeast of the study area. In addition, the human disturbance index changed greatly in the area along the river or around the cities and towns. Meanwhile, through the analysis of the quadratic regression equation, we found that there was an inverted U-shaped relationship between the human disturbance index and different landscape indices. The use of cultivated land and construction land as well as the distance to the city center, distance to the port, and distance to the industrial land affected the landscape change of the wetlands.. Attention should be paid to coordinating the development with ecological protection in the process of land use in this area, and ecological remediation should be carried out to improve the structure and ecological function of the wetlands near the city center, ports, and industrial land. The findings of the present study can provide important references for wetland protection as well as the restoration and maintenance of the sustainable development of wetland ecosystem services.

Although this study achieved its objectives, it also had some limitations. First, landscape evolution and human disturbance intensity are related to the data precision, and this study was analyzed directly based on 30 m resolution data without considering the impact of different precision on the results of landscape evolution and human disturbance intensity. Second, the study was conducted based on the data of Guannan County and Guanyun County, so further studies with different scales and regions need to be conducted to examine whether the relationship between the wetlands' landscape and human disturbance provides generally applicable results. Additionally, the accuracy of LULC data interpretation may also increase the uncertainty of the results.

**Supplementary Materials:** The following supporting information can be downloaded at: https://www.mdpi.com/article/10.3390/rs14205160/s1, Table S1: Remote sensing interpretation marks for land use/land cover in the study area.

**Author Contributions:** Conceptualization, B.W.; Methodology, J.P. and C.X.; Software, J.P.; Formal analysis, C.X.; Data curation, W.L.; Writing—original draft preparation, S.S.; Writing—review and editing, B.W. and Y.W.; Funding acquisition, B.W. and Y.W. All authors have read and agreed to the published version of the manuscript.

**Funding:** This research was funded by the National Natural Science Foundation of China (grant number 72003090) and the Fund of the Jiangsu Land Science Society (grant number JSTDXHKT202102).

**Institutional Review Board Statement:** Not applicable.

**Informed Consent Statement:** Not applicable.

**Data Availability Statement:** Not applicable.

**Conflicts of Interest:** The authors declare no conflict of interest.

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
