# Peer review of "Effects of Human Disturbance on Riparian Wetland Landscape Pattern in a Coastal Region"

_remotesensing, doi:10.3390/rs14205160_

Round 1

Reviewer 1 Report (Previous Reviewer 3)

              The revised manuscript has been improved; however, the revised manuscript needs further revision.

              Authors cited some literatures showing the validity of the authors' new model including quadratic term. However, more exact explanation showing the validity of the authors' model is necessary. If authors never have the originality of the model, authors should show what is the originality of the authors' investigation.

              Statistical analysis is obviously necessary on the data appearing in the Figure 4 and Figure 6 (Figure 5 of the original manuscript). Authors can select the appropriate statistical method and show the significance levels in each figure. I am sorry but I cannot understand the meaning of the tables (I cannot find any significance level in the tables) included in the 'response to the comments'.

              I encourage authors to make further revision.

Author Response

Q1: Authors cited some literatures showing the validity of the authors' new model including quadratic term. However, more exact explanation showing the validity of the authors' model is necessary. If authors never have the originality of the model, authors should show what is the originality of the authors' investigation.

Answer: Thank you for your kind suggestion. Based on the previous study, the nonlinear relationships between the human disturbance and the wetland landscape could be observed by the correlation analysis but ignored [1]. As the quadratic regression has been used to analyze the nonlinear characteristics in the fields of environmental conservation, energy consumption, and driving forces of land use change [2-5], the present study attempts to introduce the quadratic regression equation to quantitatively investigate the impact of human activities on the landscape pattern. In the Table 11, the regression results shows that the quadratic regression equation is effective in studying the influencing factors of wetland landscape patterns. The related contents have been highlighted in red in the revised manuscript.

References

  1. Cui, L.L.; Li, G.S.; Chen, Y.H.; Li, L.J. Response of landscape evolution to human disturbances in the coastal wetlands in northern Jiangsu Province, China. Remote Sens 2021, 13, doi:10.3390/rs13112030.
  2. Xu, B.J.; Zhong, R.Y.; Hochman, G.; Dong, K.Y. The environmental consequences of fossil fuels in China: National and regional perspectives. Sustain Dev 2019, 27, 826-837, doi:10.1002/sd.1943.
  3. Yao, S.; Zhang, S.; Zhang, X. Renewable energy, carbon emission and economic growth: A revised environmental Kuznets Curve perspective. J Clean Prod 2019, 235, 1338-1352, doi:https://doi.org/10.1016/j.jclepro.2019.07.069.
  4. Eastman, J.R.; He, J.N. A regression-based procedure for Markov transition probability estimation in land change modeling. Land 2020, 9, 12, doi:10.3390/land9110407.
  5. Li, F.; Zhang, S.W.; Bu, K.; Yang, J.C.; Wang, Q.; Chang, L.P. The relationships between land use change and demographic dynamics in western Jilin province. Geogr. Sci. 2015, 25, 617-636, doi:10.1007/s11442-015-1191-x.

Q2: Statistical analysis is obviously necessary on the data appearing in the Figure 4 and Figure 6 (Figure 5 of the original manuscript). Authors can select the appropriate statistical method and show the significance levels in each figure. I am sorry but I cannot understand the meaning of the tables (I cannot find any significance level in the tables) included in the 'response to the comments'.

Answer: The number of patches, patch density, edge density, interspersion and juxtaposition index, patch cohesion index, human disturbance index, and area in different disturbed levels were calculated respectively. For each index, only 1 value in each year. For example, in Figure 6(a), the number of beach patches was 1913, which was an accurate value. We wanted to observe the changes in different indexes among the years 2009, 2014, and 2020. We searched the relevant studies and consulted experts in statistics, statistical methods such as independent sample t-tests or analysis of variance, which are used to compare differences between two or more groups, are not suitable for analysis.

Reviewer 2 Report (Previous Reviewer 4)

I agree with the final version.

My observations are only associated with text structure and presentation, but I think that will be reviewed in the final edition. Some line spaces and table and figure format must be adjusted.

Author Response

Q1: My observations are only associated with text structure and presentation, but I think that will be reviewed in the final edition. Some line spaces and table and figure format must be adjusted.

Answer: Thanks for your kind comment. We checked the line spaces and table and figure format carefully and adjusted it.

Reviewer 3 Report (New Reviewer)

This study quantitatively analyzed the spatial and temporal characteristics of the evolution of wetland landscape patterns in the study area from 2009 to 2020 and their relationship with human disturbance, which is rich in content and reliable in results, but the content needs further revision.

1.        The importance of the study area wetlands is less described in the introduction. Why is it important to study this region? Please explain the ecological significance of the study of this region.

2.        Please pay attention to grammar, spelling, and sentence structure of manuscript so that the goals and results of the study are clear to the reader. Such as this sentence “The connectivity of landscape pattern reduced landscape diversity decreased” in Figure 9.

3.        The color of each type of the line graph in Figure 4 is similar, it is best to replace the color with a clear contrast or echo the color of the type of Figure 2.

4.        Sankey diagrams are proposed to represent land use transfer, and Tables 6, 7 and 8 can be used as supplementary data.

5.        More mechanism explanations should be added to further explain the spatiotemporal influence of driving factors on the change of landscape type.

6.        The format of many references in the manuscript is not standardized, please check and revise them one by one.

7.        A paragraph of limitation discussion should be added to clarify the limitation or uncertainty of data and methods in this current study.

Author Response

The reply is in the attachment.

Reviewer 4 Report (New Reviewer)

There is an extensive literature on the use of remote sensing to assess the status of wetlands.  As the authors correctly note, there is a need to continue those studies as globally wetlands are decreasing.  This is a very complex and lengthy study of methods to examine the changes in wetlands over time and potential causes.  The manuscript would benefit from simplification and decreased length.

There is a clear statement of intent and a logical organization to the manuscript.  The tables and figures are informative but also complex, could be reduced in number and also simplified.  At least one does not appear to be cited in the text.  There is an extensive set of appropriate references with some minor inconsistencies in format.  There are article titles in both upper and lower case and some journals are abbreviated, most are not.  The use of more than one study site would determine if the results are data/site specific or more general, however given the complexity of this study, difficult.

 The entire study is based upon spatial information extracted from remote sensing imagery.  The process of this extraction must be included in the manuscript to provide validity in the resulting extensive analysis.  For example on line 128 there needs to be a clarification as to how classes were extracted and which classes were extracted and their definitions.  Much of the data analysis is on different types of wetlands and spectrally it seems unlikely they could be accurately determined by most imagery analysis methods.  It is also unclear how river and ponds are separated from the imagery assuming that is the source of this information.  The authors state an 85% accuracy but do not specify the source and amount of validation data and if that accuracy was for one or all three years of analysis.  At least one error matrix needs to be included.  

 Similarly, the information on lines 130-135 needs clarification.  How were these decisions accomplished and what were their accuracies.  The same information is needed for the stated delineations on line 136 including from one data set these features were extracted via ArcGIS?  One potential significant concern which needs to be discussed by the authors is how does the difference in months of imagery acquisition, April to July, impacts the accuracies.  Anniversary date imagery for change detection is a standard requirement.  As a minor point, many of the surface features identified as land uses are really land covers.  The term Land Use/Land Cover (LULC) is more standard. 

 The manuscript also requires editorial review especially in the earlier sections.  A few comments follow only from the abstract:

1.       The title is unnecessarily wordy.

2.       Line 11, a river for the river.

3.       Line 13, a ecosystem for the ecosystem.

4.       Line 14, China?

5.       Line 17, is quantitatively necessary?

6.       Line 20, ; or .?

7.       Line 26, gathered is a poor work choice.

8.       It is an unusual selection and mix of keywords.

9.       There are a few examples on inconsistent use of serial commas.

10.   There are acronyms which need to be defined.

 As stated, the topic of wetland change is important.  However, this is a very complex analysis which would benefit from simplification.  It is a necessary requirement for the authors to fully document the source of their data to justify the discussion and conclusions.  

Author Response

Reviewer #4

Q1: There is a clear statement of intent and a logical organization to the manuscript. The tables and figures are informative but also complex, could be reduced in number and also simplified. At least one does not appear to be cited in the text. There is an extensive set of appropriate references with some minor inconsistencies in format. There are article titles in both upper and lower case and some journals are abbreviated, most are not.

Answer: Thanks for your kind comment. After carefully checked and revised, all the tables and figures are cited in the text. The format of references has been checked and revised one by one.

Q2: The entire study is based upon spatial information extracted from remote sensing imagery. The process of this extraction must be included in the manuscript to provide validity in the resulting extensive analysis. For example, on line 128 there needs to be a clarification as to how classes were extracted and which classes were extracted and their definitions. Much of the data analysis is on different types of wetlands and spectrally it seems unlikely they could be accurately determined by most imagery analysis methods. It is also unclear how river and ponds are separated from the imagery assuming that is the source of this information. The authors state an 85% accuracy but do not specify the source and amount of validation data and if that accuracy was for one or all three years of analysis. At least one error matrix needs to be included.

Answer: Thanks for the kind suggestion. In the present study, the land use/land cover (LULC) data for 2009,2014 and 2020 were obtained from Landsat TM images from 2009, and Landsat OLI images from 2014 and 2020 (Table 1). We selected April to July as the study period due to the images of this time having few clouds and the differences in vegetation coverage being small, which improved the classification accuracy. Before LULC data extraction, the remote sensing image preprocessing including geometric, topographic, and radiometric corrections was performed using the ENVI 5.3 software According to the Chinese National standard Current land use classification (GB/T 21010–2017) and the previous studies [6,11,13], 14 types of LULC types were classified, including ditch, river, pond, beach, paddy field, dryland, other arable land, brine pan, construction land, garden, grassland, other unused lands, traffic land, and woodland. The clarification as to how classes were extracted and which classes were extracted and their definitions were summarized in Table S1, which was uploaded as the Supplementary Materials.

For the data accuracy, random precision evaluation points were created, we selected 2000 verify grids, and after field verification and the high-resolution remote sensing image test, the accuracy of the data interpretation for each year was more than 85%, which met the precision requirement of the study.

Q3: Similarly, the information on lines 130-135 needs clarification. How were these decisions accomplished and what were their accuracies. The same information is needed for the stated delineations on line 136 including from one data set these features were extracted via ArcGIS? One potential significant concern which needs to be discussed by the authors is how does the difference in months of imagery acquisition, April to July, impacts the accuracies. Anniversary date imagery for change detection is a standard requirement. As a minor point, many of the surface features identified as land uses are really land covers. The term Land Use/Land Cover (LULC) is more standard.

Answer: Based on the image data and land use status data, the boundaries of the city, village, port, industrial land, and coastline were extracted by ArcGIS 10.3. We selected April to July as the study period due to the images of this time having few clouds and the differences in vegetation coverage being small, which improved the classification accuracy. In addition, the statement about “land use” was changed to “LULC” in the appropriate part of the study. The related contents have been supplemented in lines 124-142 and highlighted in red.

Q4: The title is unnecessarily wordy.

Answer: The title has been changed to “Effects of human disturbances on riparian wetlands’ landscape pattern in coastal region”.

Q5: Line 11, a river for the river.

Answer: The statement about “the river” was changed to “a river”.

Q6: Line 13, a ecosystem for the ecosystem.

Answer: The statement about “the ecosystem” was changed to “an ecosystem”.

Q7: Line 14, China?

Answer: A relevant supplement was made in the study.

Q8: Line 17, is quantitatively necessary?

Answer: The word was removed in the study.

Q9: Line 20, ; or .?

Answer: The “.” was changed to “;”.

Q10: Line 26, gathered is a poor work choice.

Answer: The word “gathered” was changed to “aggregated”.

Q11: It is an unusual selection and mix of keywords.

Answer: As other reviewer suggested that the keywords should be different from the words in the title, the wetlands evolution, human effects, driving forces, nonlinear relation, and quadratic regression equation were used for the keywords.

Q12: There are a few examples on inconsistent use of serial commas.

Answer: Thanks for the kind suggestion. The relevant use of the commas has been checked and modified.

Q13: There are acronyms which need to be defined.

Answer: Thanks for the kind suggestion. The abbreviation was checked and modified

Q14: As stated, the topic of wetland change is important. However, this is a very complex analysis which would benefit from simplification. It is a necessary requirement for the authors to fully document the source of their data to justify the discussion and conclusions.

Answer: Thanks for the kind suggestion. The study was rich in content. We tried to simplify it, but the reduction cannot completely express the opinion of this study. Additionally, the journal also requires a minimum of 18 pages of the study.

Round 2

Reviewer 1 Report (Previous Reviewer 3)

I suggest authors to make statistical analysis based on each pixel.

Author Response

Thanks for your kind suggestion. We searched the relevant studies and consulted experts in statistics and tried our best to deal with this problem. The response is in the attachment. We hope you can be satisfied with our explanation.

Reviewer 3 Report (New Reviewer)

The paper has certain novelty and advantages for this field research work, and has value for publishing in Journal of Remote Sensing.

Author Response

Thanks for your helpful comments. Your comments have helped us to improve our manuscript.

This manuscript is a resubmission of an earlier submission. The following is a list of the peer review reports and author responses from that submission.

Round 1

Reviewer 1 Report

It is meaningful to study the nonlinear effects of human activities on wetland landscape change. This paper uses coastal riverine cities in East China as the study area to explore the relationship between human disturbance and wetland landscape patterns, and draws novel conclusions, but more theoretical and case support is needed for this conclusion. The current version of the paper is poorly organized and needs more work to satisfy readers before it is accepted for publication. Here are the detailed comments. 

1. The conclusion is not well argued. The innovation of this study is to provide a novel conclusion that there is an inverted U-shaped relationship between human disturbance index and different landscape indices. In lines 463-466, it is mentioned that "moderate human disturbance can contribute to the protection of wetlands due to the relationship of inverted U type".

(1) The text directly supports this conclusion mainly in lines 448-450 and in Figure 9. Some of the landscape pattern indices of the wetlands show a trend of increasing and then decreasing with increasing human disturbance. Why can this represent that moderate human disturbance can promote wetland conservation? Are there sufficient theories and cases to support it?

(2) If this conclusion is valid, is there a definition of "moderate human interference" in the text, and does this "moderate" vary by study area?

2. The work of literature research is not sufficient. The details are as follows.

(1) Writing about the importance of studying coastal wetlands (lines 35-71) is too lengthy and is inappropriate. This section needs to be streamlined and to highlight the role of the specific research issue in the study of coastal wetlands.

(2) Most importantly, it is inadequate to summarize the current state of research (lines 72-90) only through the literature [19]. The logical connection between the current state of research and the scientific issue of this paper is not tight.

3. The scientific soundness of the experiment needs to be improved. The details are as follows.

(1) As seen in Figure 1, Jiangsu Province is a coastal province with a long coastline. This study focuses on coastal wetlands along the coastal river area, so why was only one small area selected as the study area? The authors need to explain the reason for the selected study area and its representative significance in the study of coastal river wetlands.

(2) The same type of wetland faces different human disturbances in urban and rural areas. Is the human disturbance index in Section 2.3.4 appropriate to assign a fixed human disturbance index for each wetland type?

(3) Section 3.1.2, this paper analyzes the landscape pattern changes at landscape level and class level (different wetland types). However, there is also spatial heterogeneity in wetland landscape patterns, and consideration should be given to enhancing the analysis of wetland landscape patterns in different locations.

(4) The results of the regression analysis in Section 3.3 (Tables 10 and 11) show that observations is 8196, is this three years of data (three times the number of grids 2732) calculated together? Is it stated in the text? Also, are there any differences in the results of the regression analysis for different years?

4. The presentation of the study data and analysis of results needs to be improved. The details are as follows.

(1) In Section 2.2, it is mentioned that the data of population and GDP are from the statistical yearbook (lines 134-136). Are the regression results in Section 3.3 derived from spatially gridded statistical yearbook data? If so, how are these data spatially gridded?

(2) In Section 2.3, it is suggested to first explain how the five methods constitute the experiments in this paper, and a flow chart can be drawn.

(3) What is the fitting accuracy of the human interference index in Figure 9 in Section 4.2? Can it be illustrated in the figure and in the text?

5. The structural arrangement of the article needs to be adjusted. The details are as follows.

(1) Section 4.1 addresses the analysis of drivers, using large amounts of literature to explain the experimental results, rather than providing an in-depth discussion of the results of this paper, and the literature research work should be placed in the introduction section. If this paper wants to analyze the drivers of wetland landscape patterns, it should be discussed through the experimental results of this paper.

(2) Section 3.1.1 only deals with changes in area, and the title "pattern" is not appropriate to summarize it. However, Section 3.1.2 deals with the analysis of landscape patterns.

Reviewer 2 Report

1. It can be seen from the figure 3 that the whole area is covered with wetlands. Therefore, is the data source reliable?

2. The size of the research grid of your formula 6 and Table 4. From Table 11, the results of this research model are not good. Is this related to the selection of indicators?

3. The variable denotes the wetland landscape pattern in the formula 6. I think it should be the wetland landscape changes.

The reasons for the change of wetland landscape is a very complex problem. The authors need to find the main reasons for the change of wetland landscape on the basis of detailed investigation, and then do this analysis.

Reviewer 3 Report

              The present paper describes the relationship between landscape change and anthropogenic disturbance in riparian wetlands by analyzing Landsat images. The topic is important for the sustainable development of riparian wetlands; however, originality of the research is not clearly described. Authors constructed the model estimating the impact of human activities on the wetland landscape including quadratic term of human disturbance index. If the model including quadratic term has originality, authors should make evaluation of the validity of the model.

              Authors describes that corresponding hemeroby index was assigned to different types of landscapes to evaluate anthropogenic disturbance on wetlands. This simply means that the index of human disturbance is dependent on landscape pattern. The index of human disturbance and the index of landscape pattern should be independently obtained. Authors should make explanation on the validity of authors' method.

              The followings are technical suggestions.

Abstract: Authors should avoid using 'northwest, northeast, and central parts', 'in the north', and 'in the south' in lines 20-22, because these are clarified after reading the main text.

Figure 4: Significance levels should be included in the figure.

Figure 5: Significance levels should be included in the figure.

Figure 6: Interval of classes (range of values with the same color) should be exactly the same among the three figures (a, b, and c). It is difficult to compare these three figures.

Figure 7: Authors should describe the correspondence between description and actual data. Evidence of data sustaining each of these descriptions are necessary.

Figure 8: Legend of figure (a) is better to be changed to other color gradation than figures b-d. Interval of classes (range of values with the same color) should be exactly the same among figures b, c, and d. It is difficult to compare these three figures.

Reviewer 4 Report

The paper presents an interesting treatment using landscape metrics and the index of human disturbance associated with statistic treatment. I think it needs only some adjustments before being accepted.

(Article) - Effects of human disturbances on wetlands’ landscape changes in riparian areas along the river in the coastal region.

(Abstract) - In the keywords, don´t repeat the title´s words.

(Introduction) - In lines 101 to 105, writing as the paper objective.

(Results) - In line 279 (Figure 3), the color scale isn´t clear to the reader. Before the figure 3 presentation, the reader needs to know the land use and land cover classification for 2009, 2014, and 2020, not only the transitions. In line 366 (Figure 6), the scale variation must be equal to compare the results.

(Discussion) - In lines 429 to 439, the driving factors and results of the evolution of wetland landscape present many factors. Are they theoretical or specific to the studied area? If theoric, I suggest changing the figure for a more contextualized approach. In line 469 (Figure 8), the scale variation must be equal to compare the results.